# Bulk flow of cerebrospinal fluid observed in periarterial spaces is not an artifact of injection

Aditya Raghunandan[1†], Antonio Ladron-de-Guevara[2†], Jeffrey Tithof[1,3], Humberto Mestre[2], Ting Du[2], Maiken Nedergaard[2,4], John H Thomas[1], Douglas H Kelley[1]*

[1]Department of Mechanical Engineering, University of Rochester, Rochester, United States; [2]Center for Translational Neuromedicine, University of Rochester Medical Center, Rochester, United States; [3]Department of Mechanical Engineering, University of Minnesota, Minneapolis, United States; [4]Center for Translational Neuromedicine, University of Copenhagen, Rochester, United States

**Abstract** Cerebrospinal fluid (CSF) flowing through periarterial spaces is integral to the brain's mechanism for clearing metabolic waste products. Experiments that track tracer particles injected into the cisterna magna (CM) of mouse brains have shown evidence of pulsatile CSF flow in perivascular spaces surrounding pial arteries, with a bulk flow in the same direction as blood flow. However, the driving mechanism remains elusive. Several studies have suggested that the bulk flow might be an artifact, driven by the injection itself. Here, we address this hypothesis with new in vivo experiments where tracer particles are injected into the CM using a dual-syringe system, with simultaneous injection and withdrawal of equal amounts of fluid. This method produces no net increase in CSF volume and no significant increase in intracranial pressure. Yet, particle-tracking reveals flows that are consistent in all respects with the flows observed in earlier experiments with single-syringe injection.

*For correspondence:
d.h.kelley@rochester.edu

[†]These authors contributed equally to this work

**Competing interests:** The authors declare that no competing interests exist.

## Introduction

Cerebrospinal fluid (CSF) flowing in perivascular spaces (PVSs) – annular tunnels that surround the brain's vasculature – plays a crucial role in clearing metabolic waste products from the brain (*Iliff et al., 2012*; *Xie et al., 2013*). The failure to remove such waste products, including toxic protein species, has been implicated in the etiology of several neurological disorders, including Alzheimer's disease (*Iliff et al., 2012*; *Peng et al., 2016*). Recently, in vivo experiments that combine two-photon microscopy and flow visualization in live mice have used the motion of fluorescent microspheres injected into the cisterna magna (CM) to measure the flow of CSF through the PVSs surrounding pial arteries. These PVSs, sometimes referred to as surface periarterial spaces, are found near the surface of the brain and are continuous with the subarachnoid space. The results show pulsatile flow, in lock-step synchrony with the cardiac cycle and with an average (bulk) flow in the same direction as that of the arterial blood flow (*Bedussi et al., 2018*; *Mestre et al., 2018b*). Characterizing the flow, however, is easier than determining its driver. Although arterial pulsation has long been considered as a possible driving mechanism for the bulk flow (*Bilston et al., 2003*; *Hadaczek et al., 2006*; *Wang and Olbricht, 2011*; *Iliff et al., 2013*; *Thomas, 2019*; *Daversin-Catty et al., 2020*), that notion remains controversial (*Diem et al., 2017*; *Kedarasetti et al., 2020a*; *van Veluw et al., 2020*), and other mechanisms are possible.

One such mechanism is the injection of tracers into the CM, which might cause a pressure gradient that drives a flow in the PVSs of pial arteries (*Smith et al., 2017*; *Smith and Verkman, 2018*;

 

*Croci et al., 2019*; *Keith Sharp et al., 2019* ; *van Veluw et al., 2020*; *Vinje et al., 2020*; *Kedarasetti et al., 2020a*; *Faghih, 2021*). Injection of CSF tracers is known to raise the intracranial pressure (ICP) by 1–3 mmHg (*Iliff et al., 2013*; *Mestre et al., 2020*), consistent with the fact that a volume of fluid is being added to the rigid skull (*Hladky and Barrand, 2018*; *Bakker et al., 2019*). If that ICP increase is not uniform, the resulting pressure gradient could drive fluid into low-resistance pathways such as PVSs surrounding pial arteries (*Faghih and Sharp, 2018*; *Bedussi et al., 2018*). In that case, the bulk flows observed in detail by *Mestre et al., 2018b* might have been artifacts of the injection. *Mestre et al., 2018b* showed that the flows did not decay over time, as would be expected if they were injection artifacts, but given that injection artifacts have been suggested in several more recent publications, we decided to test the hypothesis with additional in vivo experiments, essentially identical to the earlier experiments (*Mestre et al., 2018b*), but employing a new particle-injection method.

The new injection protocol, illustrated in *Figure 1b*, employs a dual-syringe system to infuse the tracer particles. In this system, two cannulae connected to synchronized syringe pumps are inserted into the CM; one line injects fluid in which the tracer particles are suspended, while the other line simultaneously withdraws an identical amount of fluid at the same volumetric flow rate. Thus, no net volume of fluid is added to the intracranial compartment, and hence we expect no significant change in ICP. We use two-photon microscopy to visualize the motion of the fluorescent tracer particles and measure the flow in the PVS of the cortical branches of the middle cerebral artery (MCA) using particle tracking velocimetry. We also simultaneously measure changes to ICP while monitoring heart and respiration rates. We compare the flow characteristics measured under the new protocol with those measured previously using the traditional single-injection protocol (*Bedussi et al., 2018*; *Mestre et al., 2018b*) (depicted in *Figure 1a*). (For this comparison, the data from *Mestre et al.,*

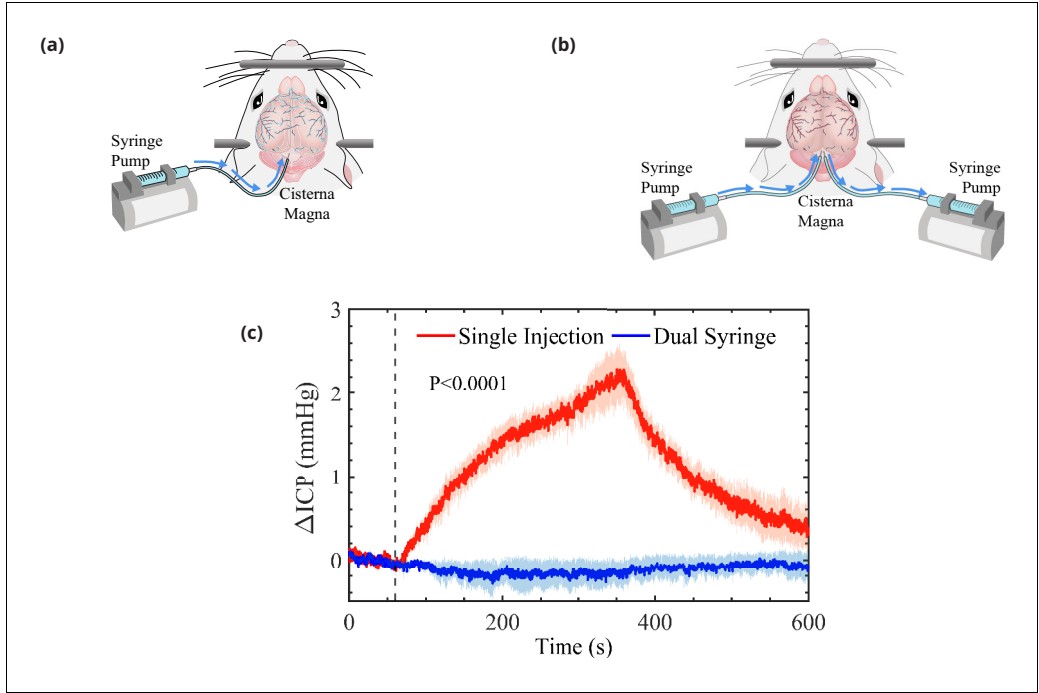

**Figure 1.** Schematic representation of the cisterna magna injection using (a) the single-injection protocol for injection of 10 μL at 2 μL/min and (b) the dual-syringe protocol for simultaneous injection and withdrawal of 20 μL at 2 μL/min. The effect of single-injection and dual-syringe tracer infusion upon intracranial pressure (ICP) is shown in (c). The ICP was monitored continuously during injection of cerebrospinal fluid (CSF) tracers into the CM of mice. Injection begins at 60 s, indicated by the vertical dashed line. Single-injection infusion of 10 μL at a rate of 2 μL/min resulted in a mild change of ~2.5 mmHg in ICP, whereas little or no change in ICP was observed during the simultaneous injection and withdrawal in the dual-syringe protocol. Repeated measures two-way analysis of variance (ANOVA) was performed; interaction p-value < 0.0001; n = 5 mice for single-injection and n = 6 mice for dual-syringe. The shaded regions above and below the plot lines indicate the standard error of the mean (SEM).

*2018b* analyzed here are from the control mice, not the hypertension mice.) Our new results are consistent in all respects with the previous results. With the new infusion protocol, the flow is again pulsatile in nature, in step with the cardiac cycle, with a net (bulk) flow in the direction of arterial blood flow. We find nearly identical mean flow speeds and other flow characteristics with the new infusion protocol. Our new experiments confirm that the flows we observed in the PVSs of pial arteries in our earlier experiments are natural, not artifacts of the tracer infusion, and provide additional statistical information about these flows.

## Results

### Changes in ICP

In a group of mice, we evaluated the effect of tracer infusion upon ICP. A 30-gauge needle was inserted stereotactically into the right lateral ventricle and connected to a pressure transducer to monitor ICP during CSF tracer injection into the CM, using both the single-injection (n = 6 mice) and dual-syringe (n = 5) protocols (*Figure 1c*). In agreement with prior studies using similar single-injection protocols (*Iliff et al., 2013*; *Xie et al., 2013*; *Mestre et al., 2020*), we found that the injection of 10 µL of CSF tracer into the CM at a rate of 2 µL/min resulted in a mild elevation of ICP (~2.5 mmHg) that relaxed to baseline values within 5 min of the cessation of injection (*Figure 1c*). When ICP was measured during the dual-syringe infusion, we observed that the simultaneous injection of the tracer and withdrawal of CSF did not significantly alter ICP (*Figure 1c*), as expected given the absence of any net change in the volume of fluid in the intracranial CSF compartment. Based on these findings, we conducted intracisternal infusion of fluorescent microspheres into the CM using the dual-syringe protocol to perform particle-tracking studies and determine the characteristics of CSF flow in the absence of any transient elevation of ICP caused by the infusion protocol.

### Flow measurements in PVSs

We studied the motion of tracer particles infused into the CM with the new dual-syringe protocol (lower panels in *Figure 2*), and compared it with the motion of tracer particles observed by *Mestre et al., 2018b* and infused with the single-injection protocol (upper panels in *Figure 2*), using particle tracking to examine flow of CSF in the PVSs of pial arteries.

The images were acquired through a sealed cranial window using intravital two-photon microscopy. The cranial window was prepared on the right anterolateral parietal bone to visualize the cortical branches of the MCA, as chosen in previous studies (*Bedussi et al., 2018*; *Mestre et al., 2018b*). In the new protocol, the particles appeared in the visualized spaces ~300 s after infusion was complete. This time scale is similar to that in our previous report (*Mestre et al., 2018b*) of 292 ± 26 s, but particle counts were lower than those observed using the single-injection technique (an average 6200 particles for the dual-syringe method vs. 19,800 particles for the single-injection method), likely because some of the injected particles were siphoned into the withdrawal line of the dual-syringe setup. However, a sufficient number of particles made their way into the PVSs to enable rigorous flow measurements (see *Figure 2—video 1*). Results obtained from the particle tracking analysis are shown in *Figure 2*. Each of the six experiments using the new protocol lasted at least 10 min. An example of the superimposed particle tracks imaged in an experiment is shown in *Figure 2e*. The particle tracks are mostly confined to the PVSs surrounding the artery, occasionally crossing from one side of the artery to the other. The distribution of particle tracks is spatially continuous across the width of the imaged PVSs under both infusion methods (*Figure 2a*; *Mestre et al., 2018b* and *Figure 2e*), reaffirming that PVSs along pial arteries are open, rather than porous, spaces (*Min Rivas et al., 2020*). The direction of the observed fluid flow in the different branches is indicated by the arrows in *Figure 2b* and *Figure 2f*. If injection were driving the flow, we would expect to observe dominant directional transport of tracer particles only when using the single-injection method, and little or no transport when using the dual-syringe method. The time-averaged (bulk) flow for both infusion methods is in the same direction as that of the blood flow, providing evidence that CSF flow in PVSs is not caused by the injection. For both infusion methods, we observed no net flow in the direction opposite to that of blood flow, as some recent reports have suggested (*Aldea et al., 2019*; *van Veluw et al., 2020*). *Figure 2g* shows that the average flow speed in the PVSs of pial arteries varies across the PVS, consistent with prior reports (*Mestre et al., 2018b*) shown in

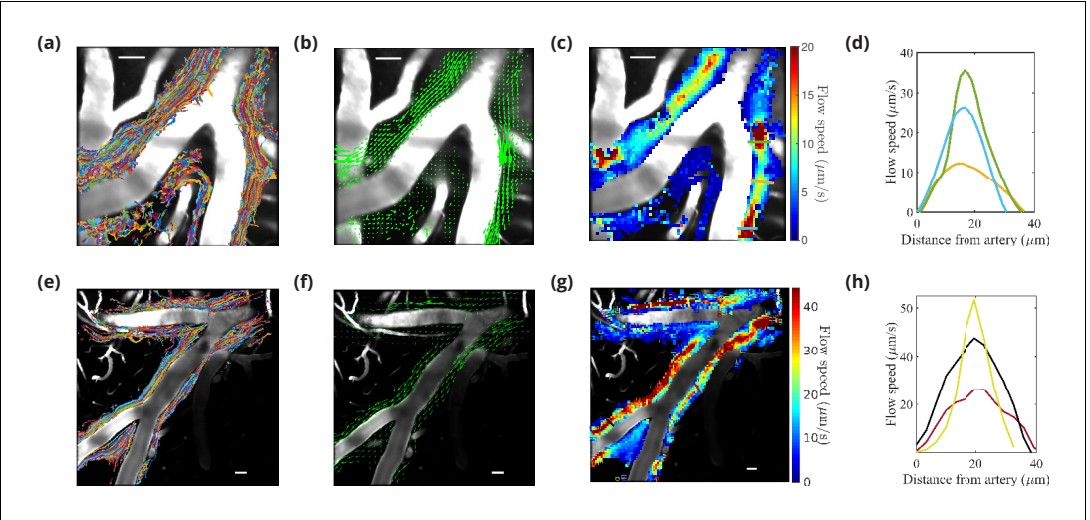

**Figure 2.** Particle tracking velocimetry in the PVSs) surrounding cortical branches of the MCA using the single-injection method (panels in first row [***Mestre et al., 2018b***]) and the new dual-syringe method (second row). The superimposed particle tracks shown in panels (**a**) and (**e**) have similar, continuous spatial distributions and show similar sizes of the perivascular spaces. The time-averaged velocity fields shown in panels (**b**) and (**f**) both show net flow of fluid in the same direction as the blood flow. The flow-speed distributions plotted in panels (**c**) and (**g**) show comparable speeds, with the fastest flow at the center of the imaged periarterial space and the slowest flow near the boundaries. Panels (**d**) and (**h**) show average flow-speed profiles across the corresponding colored lines spanning the PVS in panels (**c**) and (**g**), smoothed by interpolation. The parabolic-like nature of these velocity profiles is what is expected for viscous flow in an open channel. Scale bars indicate 50 μm. Figure panels (**a, b, c, and d**) reproduced from ***Figure 1***, ***Mestre et al., 2018a***, *Nature Communications*, published under the Creative Commons Attribution 4.0 International Public License (CC BY 4.0; https://creativecommons.org/licenses/by/4.0/). © 2018, Mestre et al. Figure panels (a), (b), (c), and (d) reproduced from ***Figure 1***, ***Mestre et al., 2018a***, Nature Communications, published under the Creative Commons Attribution 4.0 International Public License.

The online version of this article includes the following video for figure 2:

**Figure 2—video 1.** Particles infused using dual-syringe method are transported downstream in pial perivascular spaces (PVSs).

https://elifesciences.org/articles/65958#fig2video1

---

*Figure 2c.* The velocity profile is parabolic-like (*Figure 2d and h*); the flow is fastest (~50 μm/s) at the center of the PVS and slows to zero at the walls. This parabolic-like shape is consistent with laminar, viscous-dominated flow of CSF through an open annular space, and not through a porous medium, indicating that pial periarterial spaces are open (*Min Rivas et al., 2020*).

Further analysis of the data obtained from particle tracking demonstrates the close similarity between the flows observed in the two protocols, as shown in *Figure 3*. A time-history of the measured flows — quantified by the spatial root-mean-square velocity computed at each instant of time ($V_{rms}$) — portrays very similar behavior over times much longer than the time it takes for the ICP to return to normal after the infusion (*Figure 3a*). (The times shown here begin when particles were first seen or when the imaging was started: these times differ by less than 1 min and so do not affect the results significantly.) The pulsatile nature of the flow at small time scales is depicted in *Figure 3b and c*. If injection-induced elevated ICP were driving the flow, we would observe large $V_{rms}$ values early in the single-injection experiments, followed by an exponential decay, and we would observe little or no flow in the dual-syringe experiments, in which the ICP remains unchanged. Since we observe very similar trends in the time-history profiles in both infusion protocols, the mechanisms driving the flow are apparently independent of the infusion method.

*Figure 3c* shows mean flow speeds computed by averaging the downstream velocity component over space and time for each experiment. The overall mean flow speed (open circles) is 15.71 ± 6.2 m/s for all the single-injection experiments and 17.67 ± 4.42 for all the dual-syringe experiments, values that differ by less than the standard error of the mean in either set of experiments. Significantly

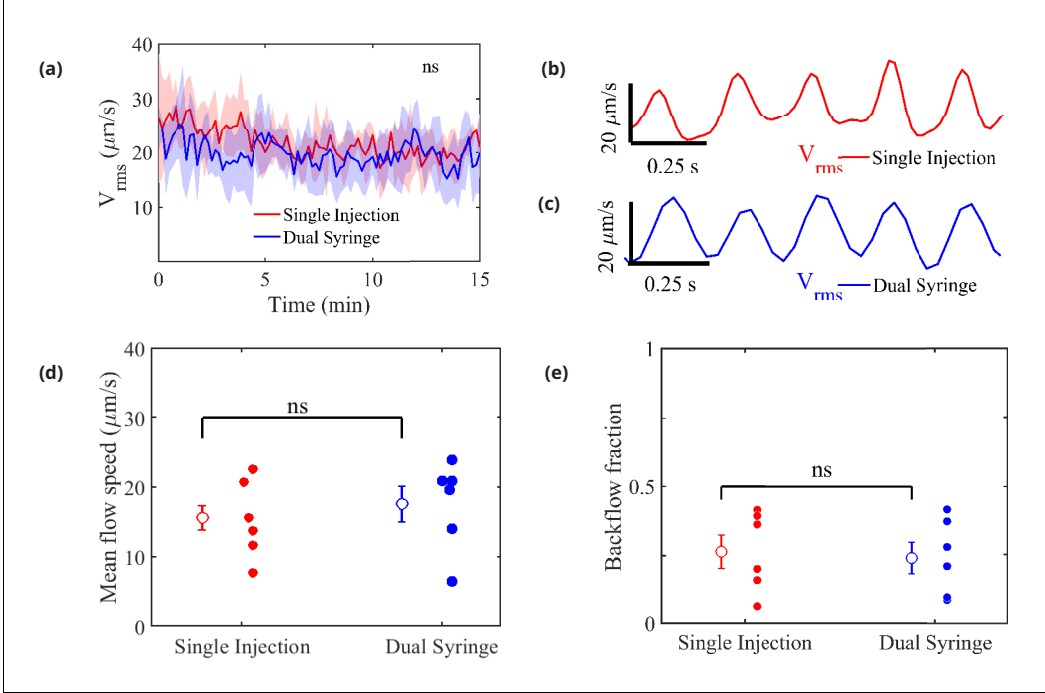

**Figure 3.** Measured flow characteristics. Panel (a) shows $V_{rms}$ over the course of the velocity measurements for both infusion methods. Repeated measures two-way ANOVA was performed; ns, not significant; n = 5 mice for single-injection and n = 6 mice for dual-syringe. The solid lines represent the mean value of $V_{rms}$ and the shaded area represents the standard error of the mean within each time bin. The pulsatility of typical measured flows is depicted in panels (b) and (c). Panel (d) shows mean downstream flow speeds and panel (e) shows backflow fractions for the individual experiments, with overall mean values shown as open circles (and bars showing the standard error of the mean). The nearly identical values for the two protocols demonstrate that the flow is independent of the injection method employed. Unpaired Student's *t*-test was performed; n = 5 or 6 mice per group; ns, not significant; mean ± SEM.

The online version of this article includes the following source data for figure 3:

**Source data 1.** Source data for panels (a, d), and (e).

greater differences in mean flow speed are caused by animal-to-animal variations than by changing from single-injection to dual-syringe methods. These values are also nearly identical to the mean speed of 17 ± 2 reported by *Bedussi et al., 2018*, from experiments that used a single-injection protocol with a lower injection rate. The mean flow speeds represent the speeds at which tracer particles (or CSF) are transported in the direction of arterial blood flow (downstream), and presumably into the brain. If the observed flows were injection-induced, we would expect faster mean flows with the single-injection method than with the dual-syringe method.

We also computed a 'backflow fraction' for each experiment, as the fraction of the downstream velocity measurements showing motion in the retrograde direction (opposite that of the blood flow): the results are shown in *Figure 3e*. An injection-driven flow would exhibit a much smaller backflow fraction. However, the backflow fraction is nearly identical: 0.26 ± 0.059 for single-injection and 0.24 ± 0.056 for dual-syringe infusion respectively. As with flow speed, mean values differ by less than the standard error of the mean, so animal-to-animal variations exceed the effects of changing the injection protocol. The nearly identical mean flow speeds and backflow fractions further demonstrate that the observed flows are natural, and not artifacts of the infusion.

## Flows pulse in synchrony with the cardiac cycle

It has been variously suggested that CSF flow might be driven by the cardiac cycle, the respiratory cycle, or perhaps both (*Rennels et al., 1985*; *Hadaczek et al., 2006*; *Yamada et al., 2013*; *Bedussi et al., 2018*), with evidence indicating much stronger correlation with the cardiac cycle

(*Iliff et al., 2013*; *Mestre et al., 2018b*). We used the simultaneous measurements of the electrocardiogram (ECG) and respiration in conjunction with particle tracking to determine the relative importance of the cardiac and respiratory cycles (*Santisakultarm et al., 2012*), and also to see if there is any difference in these relationships between the two infusion methods (*Figure 4*).

We find that the measured time-dependent components of flow quantities such as $V_{rms}$ are strongly modulated by the cardiac cycle but only weakly by respiration (*Figure 4a*; *Mestre et al., 2018b* and *Figure 4b*). This strong correlation of the pulsatile component of flow with the heart rate is exhibited under both the single-injection and dual-syringe protocols, as shown in *Figure 4c and e*, where the peak in the $V_{rms}$ occurs soon after the peak in the cardiac cycle. Probability density functions of $\Delta t$, the delay time between peaks in $V_{rms}$ and cardiac/respiratory cycles, also predict a much greater likelihood of peaks in $V_{rms}$ following the peak in the cardiac cycle (*Figure 4g* and *Figure 4h*). We observe nearly identical average delay times of ~0.05 s between peaks in $V_{rms}$ and the cardiac cycle for both protocols (*Figure 4i*). No such correlation is observed when the $V_{rms}$ is conditionally averaged over respiration cycles (*Figure 4d and f*). These observations corroborate prior reports

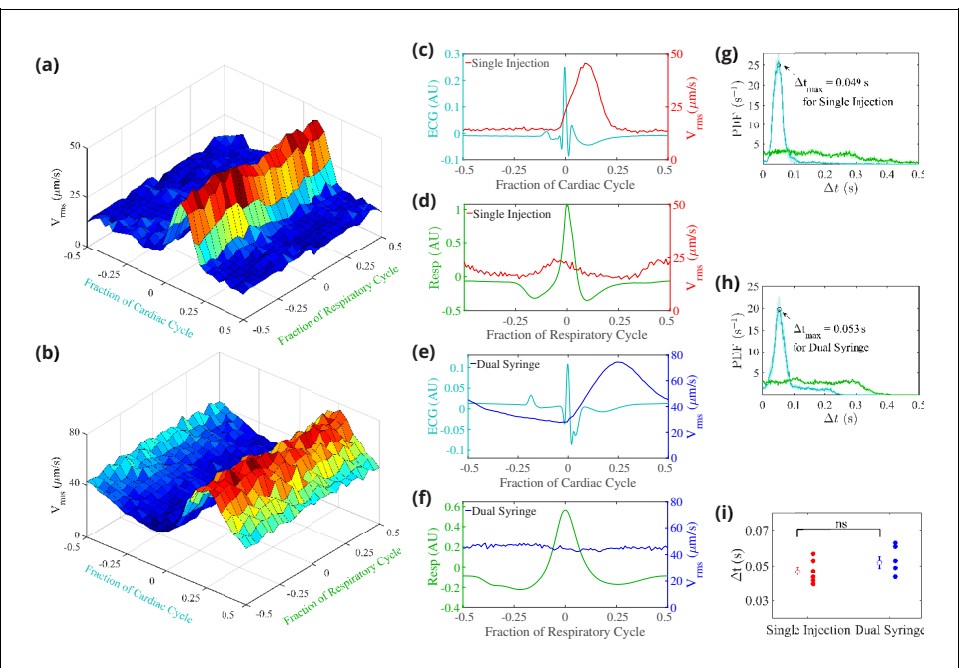

**Figure 4.** Cerebrospinal fluid (CSF) velocity variations over the cardiac and respiratory cycles. Panels (**a**) and (**b**) show the measured $V_{rms}$ conditionally averaged over the cardiac and respiratory cycles, based on the synchronized measurements of ECG, respiration, and velocity, for the single-injection (**a**) and the dual-syringe (**b**) protocols. Panel (**c**) for single injection and panel (**e**) for dual syringe both show that the peaks in the ECG are immediately followed by peaks in $V_{rms}$, indicating a strong correlation between heart rate and fluid motion in both injection protocols. No consistent trends are seen when $V_{rms}$ is averaged over the respiratory cycle, as shown in panels (**d**) and (**f**). Panels (**g**) and (**h**) show the mean and the standard error of the mean of probability density functions of the delay time $\Delta t$ between the peak in the cardiac (cyan) or respiration (green) cycle and the subsequent peak in $V_{rms}$, for single-injection (n = 5) and dual-syringe (n = 6) methods respectively. Panel (**i**) shows the average $\Delta t$ between peaks in the cardiac cycle and $V_{rms}$ for both protocols; in both, the peak in $V_{rms}$ typically occurs ~0.05 s after the peak in the cardiac cycle. Unpaired Student's *t*-test was performed; n = 5 or 6 mice per group; ns, not significant; mean ± SEM. Figure panel (**a**) reproduced from Figure 3 *Mestre et al., 2018b*, *Nature Communications*, published under the Creative Commons Attribution 4.0 International Public License (CC BY 4.0; https://creativecommons.org/licenses/by/4.0/).

The online version of this article includes the following source data for figure 4:

**Source data 1.** Data for panel (**i**).

that the cardiac cycle drives the dominant oscillatory component of CSF flow in the PVSs of pial arteries, unaffected by injection protocol.

## Discussion

Healthy removal of metabolic waste from the brain is believed to occur via circulation of CSF, which enters brain tissue through the PVSs surrounding pial arteries (*Rasmussen et al., 2018*; *Reeves et al., 2020*; *Nedergaard and Goldman, 2020*). Whereas experiments in live mice have shown that fluid is pumped in the direction of blood flow and into brain, perhaps by forces linked to the pulsation of arterial walls, several published papers have hypothesized that the observed flows might instead be artifacts of non-natural elevation of ICP caused by tracer infusion into the CM (*Smith et al., 2017*; *Smith and Verkman, 2018*; *Croci et al., 2019*; *Keith Sharp et al., 2019*; *van Veluw et al., 2020*; *Vinje et al., 2020*; *Kedarasetti et al., 2020a*; *Faghih, 2021*). In this study, we designed a new infusion protocol that enabled tracer-particle infusion with no net addition of fluid and near-zero changes in ICP. Using two-photon microscopy and particle-tracking velocimetry, we found flows of CSF in the PVSs of the cortical branches of the MCA that are statistically identical to the flows found earlier in the same location using the single-injection protocol (*Mestre et al., 2018a*). These findings are consistent with the hypothesis that the observed flows in pial PVSs are not driven by tracer infusion.

Further support for the hypothesis that these flows are not driven by tracer injection comes from their timing. The time that elapses between beginning injecting particles and observing them in pial PVSs is similar with either protocol. If injection were driving flow, we would expect particles to arrive in PVSs more quickly with the single-injection protocol than with the dual-syringe protocol. Moreover, in single-injection experiments, ICP returns to its baseline value within 5 min after injection is complete (*Figure 1*), but flows continue through the duration of the experiments (1030 min), which would not occur if ICP were the driver. By the same reasoning, if ICP were the driver, flows would not occur at all in dual-syringe experiments, but we again observe flows continuing through the duration of the experiments.

Other characteristics of the flows we observe also support the hypothesis that flows are not driven by tracer injection. If elevated ICP levels create large pressure differences across the brain, these pressure differences should undergo exponential decay because of the brain's compliance and proclivity to achieve stasis. If this exponential relaxation of ICP were to drive fluid flow, the measurements from particle tracking would reflect this decay, exhibiting fast flows at early times which then gradually subside. However, our measurements show that the mean flow remains nearly constant and similar over periods that are two to three times longer than the infusion time, for several healthy mice and both infusion protocols (*Figure 3a*). Variation between the two protocols is similar to animal-to-animal variation. In dual-syringe experiments, we observe flow speeds that are nearly identical to those observed in one earlier, single-injection study (*Mestre et al., 2018b*), and very close to those in a single-injection study that used injection rates one order of magnitude smaller (*Bedussi et al., 2018*), again suggesting that injection rate is not the driver. Finally, if the ICP elevation induced by the single-injection protocol were responsible for the tracer penetration into the brain, then variations associated with arousal state (*Xie et al., 2013*), anesthesia (*Hablitz et al., 2019*), blood pressure (*Mestre et al., 2018a*), and other biological mechanisms would not have occurred.

Though we were able to characterize flow in pial PVSs in great detail using particle tracking, quantifying flow in more distal PVSs, such as those surrounding penetrating arteries and arterioles (including Virchow-Robin spaces), remains a challenge. We found that particles large enough to be tracked individually are apparently sieved and are not transported into such smaller PVSs, consistent with prior studies (*Mestre et al., 2018a*; *Bedussi et al., 2018*). Additionally, the limitations in temporal resolution of the current microscopes in the third dimension (z-direction) prevent accurate measurement of CSF flow in PVSs of penetrating arteries and arterioles, which are oriented along that dimension. However, conservation of mass implies that fluid flowing through pial PVSs must continue through whatever regions are contiguous, and we hypothesize that Virchow-Robin spaces are contiguous. That hypothesis, which we hope to test directly in future work, is supported by data collected using several complementary methodologies in mice (*Iliff et al., 2012*; *Mortensen et al., 2019*; *Koundal et al., 2020*) and by recent MRI studies in humans (*Ringstad et al., 2017*;

*Ringstad et al., 2018*), where tracer (if not particles) is carried into the parenchyma. If that hypothesis holds, it would imply that flow in PVSs surrounding penetrating arterioles (including Virchow-Robin spaces) is not an artifact of injection, either.

Single-injection experiments do introduce substantial additional fluid into the subarachnoid space. Recent studies have infused 10 μL at rates of 1–2 μL/min (*Bedussi et al., 2018*; *Mestre et al., 2018b*), much greater than the natural CSF production rate (0.1–0.3 μL/min [*Rudick et al., 1982*; *Liu et al., 2020*]). Given the measurements and reasoning described above, it seems that little or none of the additional fluid flows through pial PVSs. Still, it must flow somewhere, and we speculate that it takes an alternate path. Recent publications have presented evidence of fluid efflux via meningeal lymphatic vessels located around the venous sinuses and at the base of the skull (*Louveau et al., 2018*; *Ahn et al., 2019*) and via pathways along cranial and spinal nerves (*Ma et al., 2019*; *Stanton et al., 2021*). Attempts to quantify the perivascular and lymphatic transport using radiolabeled tracers and contrast agents for T1 mapping techniques have estimated that around 20% of the tracers injected into the CM is transported into the brain, implying that most of the tracer mass migrates to spine and/or drains into the different efflux routes (*Iliff et al., 2012*; *Lee et al., 2018*; *Eide et al., 2018*). In addition, other studies have reported a correlation between the rate of CSF absorption and the volume of the injection or its effect upon ICP (*Boulton et al., 1998*; *Johnston and Papaiconomou, 2002*; *Ma et al., 2019*). The exact routes of these alternate paths, and the conditions under which they carry substantial amounts of fluid, would both be promising topics for future study. Moreover, since flow in pial PVSs is hardly affected by the injection of large amounts of fluid that passes along alternate paths, the mechanisms driving CSF through pial PVSs may be nearly independent of the mechanisms driving CSF flows elsewhere. Additional study might elucidate the situation.

Our results confirm that the cardiac cycle — not respiration — drives the oscillatory component of the observed flows in the PVSs of pial arteries. The peaks of $V_{rms}$ that we measured across specimens, for both infusion protocols, appear shortly after the peaks in the cardiac cycle (*Figure 4c and e*), but are not correlated with the respiratory cycle (*Figure 4d and f*). Probability density functions show that the delay times between the peaks in the ECG and the peaks in $V_{rms}$ are nearly identical for the two infusion methods (*Figure 4g and h*). Although we present compelling evidence that the cardiac cycle drives the purely oscillatory component of the pulsatile flow in pial PVSs, we cannot rule out other natural mechanisms that might be driving the average (bulk) flow, such as CSF production, functional hyperemia (*Kedarasetti et al., 2020b*), or vasomotion (*van Veluw et al., 2020*). While our results show that respiration is not a dominant driving force of flow in pial PVSs, the proximal segments of the influx routes along which CSF is transported into the brain, respiration might yet contribute to flow when CSF exits the brain (*Kiviniemi et al., 2016*). We do conclude, however, that the currently employed methods of tracer infusion are not responsible for the observed flows in pial PVSs.

The suggested role of arterial pulsatility in perivascular transport and the changes in the properties of the arterial wall in cardiovascular diseases may provide a causal linkage between vascular disorders and protein-aggregation disorders (*Nedergaard and Goldman, 2020*). For example, hypertension, an established risk factor for Alzheimer's Disease (correlated with amyloid-β accumulation [*Gentile et al., 2009*; *Carnevale et al., 2016*]), causes stiffening of the arterial wall, reducing its compliance and pulsatility, and thereby attenuating perivascular flow (*Mestre et al., 2018a*; *Mortensen et al., 2019*). These accumulating data suggest that dysfunction in perivascular transport may be one of the key causes of protein mis-aggregation and may provide a novel therapeutic approach to the prevention and treatment of neurodegenerative disease (*Nedergaard and Goldman, 2020*). Our results also have implications for drug delivery to the CNS by bolus intrathecal administration of drugs directly to the CSF. Since we have demonstrated that the methods of tracer injection do not directly affect the measured bulk flows in PVSs, the manipulation of perivascular transport rather than the volume or rate of the bolus intrathecal injection may represent a novel strategy for improving the CNS delivery of intrathecally administered drugs bypassing the blood–brain barriers (*Plog et al., 2018*; *Lilius et al., 2019*). An improved understanding of the mechanisms that drive CSF flow in the brain remains an important topic for future work.

The dual-syringe infusion method we have developed avoids increases to the volume of fluid in the skull and does not increase the ICP (*Figure 1*). Thus, the dual-syringe infusion causes minimal flow and is ideally suited to test whether injection-induced pressure differences could propel the

observed periarterial CSF flows. However, implanting a second cannula to withdraw fluid and avoid transient changes to ICP lowers the count of tracer particles injected, and would likewise reduce the amount of fluorescent dye tracer injected. Dual-syringe infusion also adds an additional step to an already complex surgery, thereby reducing the success rate with mouse specimens. The results and reasoning described above support the conclusion that flows in pial periarterial spaces, observed in the present study and in prior efforts that used the single-injection method to infuse tracer (*Iliff et al., 2012*; *Xie et al., 2013*; *Bedussi et al., 2018*; *Mestre et al., 2018b*), are affected very little or not at all by changing between these infusion methods. Thus, the dual-syringe method is not necessary for future studies that adopt prevalent infusion methods to inject tracer particles or fluorescent dyes and study flows in these proximal CSF influx pathways. However, it may be useful when validating whether new injection parameters affect CSF flow or assessing if flows in other anatomical regions are affected by the addition of extra volume or the transient changes in ICP.

## Materials and methods

### Key resources table

| Reagent type (species) or resource | Designation | Source or reference | Identifiers | Additional information |
|---|---|---|---|---|
| Strain strain background (*Mus musculus*, C57BL/6NCrl) | Male wild type (WT) | Charles River, 027 | RRID: IMSR_CRL:27 | |
| Software, algorithm | MATLAB | Mathworks | RRID: SCR001622 | |
| Software, algorithm | GraphPad Prism 8 | GraphPad Software | RRID: SCR002798 | |
| Other | FluoSpheres | Invitrogen | Cat: #13081 | Particles are 1.0 µm in diameter with excitation/emission at 580/605 nm |
| Other | Artificial CSF | Sigma-Aldrich | | Components in aCSF (concentrations in mM): 126.0 NaCl 3 KCl, 2 MgSO$_4$, 10.0 dextrose, 26.0 NaHCO$_3$, 1.25 NaH$_2$PO$_4$, 2 CaCl$_2$ |

### Animals and surgical preparation

All experiments were approved and conducted in accordance with the relevant guidelines and regulations stipulated by the University Committee on Animal Resources of the University of Rochester Medical Center (Protocol No. 2011-023), certified by Association for Assessment and Accreditation of Laboratory Animal Care. An effort was made to minimize the number of animals used. We used 8- to 12-week-old male C57BL/6 mice acquired from Charles River Laboratories (Wilmington, MA, USA). In all experiments, animals were anesthetized with a combination of ketamine (100 mg/kg) and xylazine (10 mg/kg) administered intraperitoneally. Depth of anesthesia was determined by the pedal reflex test. Once reflexes had ceased, anesthetized mice were fixed in a stereotaxic frame for the surgical procedure, and body temperature was kept at 37°C with a temperature-controlled warming pad.

### Dual-syringe protocol

For in vivo imaging, anesthetized mice were fixed in a stereotaxic frame and body temperature was maintained at 37.5°C with a rectal probe-controlled heated platform (Harvard Apparatus). A cranial window was prepared over the right MCA distribution. The dura was left intact, and the craniotomy ($\simeq 4\,\mathrm{mm}$ in diameter) was filled with aCSF, covered with a modified glass coverslip, and sealed with dental acrylic. Afterwards, two 30-gauge needles were inserted into the CM, as previously described (*Xavier et al., 2018*). Briefly, the dura mater of mice was exposed after blunt dissection of the neck muscles so that a cannula could be implanted into the CM, which is continuous with the

subarachnoid space. Using a syringe pump (Harvard Apparatus Pump 11 Elite), red fluorescent polystyrene microspheres (FluoSpheres 1.0 μm, 580/605 nm, 0.25% solids in aCSF, Invitrogen) were infused up to a total volume of 20 μL via one of the CM cannulae while CSF was simultaneously withdrawn through the other cannula at an equal rate of 2 μL/min with a coupled syringe pump.

## ICP measurements

Anesthetized mice were fixed in a stereotaxic frame, and two 30-gauge needles were inserted into the CM, as described above. A third cannula was inserted via a small burr hole into the right lateral ventricle (0.85 mm lateral, 2.10 mm ventral, and 0.22 mm caudal to bregma). Mice were then placed in a prone position. In the first set of experiments, 10 μL of artificial CSF (aCSF) was injected into the CM at a rate of 2 μL/min via one of the CM cannulae using a syringe pump (Harvard Apparatus Pump 11 Elite). In the second set of experiments, aCSF was injected at the same rate while withdrawing CSF from the CM via the other CM cannula at an equal rate using a coupled syringe pump (Harvard Apparatus Pump 11 Elite). In both experiments, ICP was monitored via the ventricle cannulation connected to a transducer and a pressure monitor (BP-1, World Precision Instruments). ICP was acquired at 1 kHz, digitized, and monitored continuously for the duration of the infusion experiments with a DigiData 1550B digitizer and AxoScope software (Axon Instruments).

## In vivo two-photon laser-scanning microscopy

Two-photon imaging was performed using a resonant scanner B scope (Thorlabs) with a Chameleon Ultra II laser (Coherent) and a 20× water immersion objective (1.0 NA, Olympus). Intravascular FITC-dextran and red microspheres were excited at a 820 nm wavelength and images were acquired at 30 Hz (ThorSync software) simultaneously with physiological recordings (ThorSync software), as previously described (*Mestre et al., 2018b*). To visualize the vasculature, fluorescein isothiocyanate–dextran (FITC–dextran, 2,000 kDa) was injected intravenously via the femoral vein immediately before imaging. Segments of the MCA were distinguished on the basis of morphology: surface arteries passing superficially to surface veins and exhibiting less branching at superficial cortical depths. ECG and respiratory rate were acquired at 1 kHz and 250 Hz, respectively, using a small-animal physiological monitoring device (Harvard Apparatus). The signals were digitized and recorded with a DigiData 1550A digitizer and AxoScope software (Axon Instruments).

## Image processing

Images with spatial dimensions 512 × 512 were obtained from two-photon microscopy. Each image is 16-bit with two channels, red and green. The FITC-dextran injected in the vasculature is captured via the green channel while the red channel is used to image the fluorescent microspheres flowing in the PVSs. Image registration via rigid translation is performed on each image in the time series to account for movement by the mouse in the background. The image registration is implemented using an efficient algorithm in MATLAB (*Guizar-Sicairos et al., 2008*) to an accuracy of 0.2 pixels. Erroneous correlations in the translation are manually corrected by linear interpolation. The translations obtained are sequentially applied to images that are padded with zero-value pixels. This ensures spatial dimension homogeneity across all images without modifying the image resolution. Particles are then detected by applying a minimum intensity threshold to each image. Typically, particles were resolved across 3–4 pixels in the image with spatial resolution of 1.29 μm.

## Particle-tracking velocimetry

The particles detected in each image were tracked using an automated PTV routine in MATLAB (*Kelley and Ouellette, 2011*; *Ouellette et al., 2006*) and can be downloaded from the repository found at *Raghunandan, 2021*. Briefly, the algorithm locates each particle with a sub-pixel accuracy and obtains a series of particle locations (particle tracks) for the entire duration of the recorded video. Particle velocities were calculated by convolution with a Gaussian smoothing and differentiation kernel. Stagnant particles that have adhered to the wall of the artery or the outer wall of the PVS, and hence no longer track the CSF flow, were masked in each image by subtracting a dynamic background image. This image was different for each frame and was computed by taking the average of 100 frames before and 100 frames after the given image. This method of masking was applied only to the dual-syringe data; the single-injection data used a simpler masking approach

with a single background image (*Mestre et al., 2018b*). Time-averaged flow velocities were obtained by segregating the imaged domain into a 70 × 70 grid, with a resolution of 7.5 × 7.5 pixels in each direction. All velocity measurements for a chosen time interval were binned based on their grid position. Average flow speeds were computed using bins with at least 15 measurements. The downstream velocity component was calculated as the dot product $u \sim \cdot \sim \hat{u}_{avg}$, where $u$ is the instantaneous particle velocity and $\hat{u}_{avg}$ is the field of unit vectors computed from the time-averaged flow field, in the direction of arterial blood flow.

## Statistical analysis

All statistical analyses were performed on GraphPad Prism 8 (GraphPad Software). Data in all graphs are plotted as mean ± standard error of the mean (SEM) over the individual data points and lines from each mouse. Statistical tests were selected after evaluating normality (D'Agostino Pearson omnibus test). When the sample size did not allow for normality testing, both parametric and non-parametric tests were performed and, in all cases, yielded the same result. Sphericity was not assumed; in all repeated measures, two-way ANOVAs and a Geisser-Greenhouse correction were performed. All hypothesis testing was two-tailed and exact p-values were calculated at a 0.05 level of significance and stated in the figure legends.

## Acknowledgements

We thank Keith Sharp for recommending something like our dual-syringe experiments to us in July 2019. We also thank Dan Xue for drafting the schematic.

## Additional information

### Funding

| Funder | Grant reference number | Author |
| --- | --- | --- |
| National Institutes of Health | RF1AG057575 | Maiken Nedergaard<br>John H Thomas<br>Douglas H Kelley<br>Aditya Raghunandan |
| Army Research Office | MURI W911NF1910280 | Antonio Ladron-de-Guevara<br>Humberto Mestre<br>Maiken Nedergaard<br>John H Thomas<br>Douglas H Kelley<br>Ting Du |
| Burroughs Wellcome Fund | Career Award at the Scientific Interface | Jeffrey Tithof |

The funders had no role in study design, data collection and interpretation, or the decision to submit the work for publication.

### Author contributions

Aditya Raghunandan, Data curation, Formal analysis, Validation, Investigation, Visualization, Writing - original draft, Writing - review and editing; Antonio Ladron-de-Guevara, Data curation, Formal analysis, Investigation, Writing - original draft, Writing - review and editing; Jeffrey Tithof, Software, Methodology, Writing - review and editing; Humberto Mestre, Methodology, Writing - review and editing; Ting Du, Methodology; Maiken Nedergaard, Supervision, Funding acquisition, Methodology, Writing - review and editing; John H Thomas, Conceptualization, Supervision, Writing - original draft, Project administration, Writing - review and editing; Douglas H Kelley, Software, Supervision, Funding acquisition, Validation, Methodology, Project administration, Writing - review and editing

### Author ORCIDs

Aditya Raghunandan ⓘ https://orcid.org/0000-0002-1938-0184
Antonio Ladron-de-Guevara ⓘ https://orcid.org/0000-0003-1093-2509

Jeffrey Tithof [iD] http://orcid.org/0000-0003-2083-0901
Humberto Mestre [iD] http://orcid.org/0000-0001-5876-5397
John H Thomas [iD] https://orcid.org/0000-0002-7127-8654
Douglas H Kelley [iD] https://orcid.org/0000-0001-9658-2954

## Ethics

Animal experimentation: All experiments were approved and conducted in accordance with the relevant guidelines and regulations stipulated by the University Committee on Animal Resources of the University of Rochester Medical Center (Protocol No. 2011-023),certifiedby Association for Assessment and Accreditation of Laboratory Animal Care.

## Decision letter and Author response

Decision letter https://doi.org/10.7554/eLife.65958.sa1
Author response https://doi.org/10.7554/eLife.65958.sa2

## Additional files

### Supplementary files

- Transparent reporting form

### Data availability

All data generated or analyzed for this study are included in the manuscript. Source data files have been provided for Figures 4a, 4d, 4e, and 5i. The particle-tracking Matlab code used in this study is available in the public domain GitLab repository found here: https://gitlab-public.circ.rochester.edu/araghuna/bulk-flow-is-not-an-artifact_raghunandan_et_al_2021.git.

The following dataset was generated:

| Author(s) | Year | Dataset title | Dataset URL | Database and Identifier |
|---|---|---|---|---|
| Raghunandan A | 2021 | Particle Tracking Raghunandan et al 2021 | https://gitlab-public.circ.rochester.edu/araghuna/bulk-flow-is-not-an-artifact_raghunandan_et_al_2021.git | Gitlab, 19aa2e2 |

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
