## [Decision Letter]

**Acceptance summary:**

We understand that your work brings an important piece of evidence for the field to understand the flow of cerebrospinal fluid observed in periarterial spaces.

**Decision letter after peer review:**

Thank you for submitting your article "Bulk flow of cerebrospinal fluid observed in periarterial spaces is not an artifact of injection" for consideration by *eLife*. Your article has been reviewed by two peer reviewers, and the evaluation has been overseen by a Reviewing Editor and Ronald Calabrese as the Senior Editor. The following individual involved in review of your submission has agreed to reveal their identity: Per K. Eide (Reviewer #1).

The reviewers have discussed the reviews with one another and the Reviewing Editor has drafted this decision to help you prepare a revised submission.

Summary:

This work addresses a timely question regarding the scientific controversy about glymphatic function. The glymphatic system was initially described in 2012 using in vivo two-photon imaging of paravascular tracer movement after injection of CSF tracers of variable size to the cisterna magna. The authors describing the glymphatic system have argued that tracer movement is convective, i.e. pressure gradient driven, created by the arterial pressure pulsations. However, the antegrade para-arterial tracer movement observed in vivo using two-photon imaging was suggested to result from an artificial increase in volume and pressure induced by the tracer infusion (about 1-3 mmHg in ICP). The present work further addresses this important methodological issue.

The authors established a methodology where artificial CSF was injected to cisterna magna via one needle and the same volume withdrawn via another needle, in order to avoid an artificial volume increase. Intracranial pressure was monitored to control for increase in ICP. By doing so, the authors robustly show that the previously reported paravascular flow is not induced by methods of tracer infusion. The study is well-designed and the results are compelling. The flow of fluid in the perivascular spaces occurs in the same direction as the blood flow and the flow appears to be driven by pulsatile blood flow.

Essential revisions:

1) One aspect that the authors need to clarify is the location for particle movement, which is referred to as the "surface periarterial spaces". This location should be better explained.

2) Do the authors refer to branches of the medial cerebral artery within the subarachnoid space and surrounded by the arachnoid membrane? I would assume it is difficult to perform measurements within the Virchow-Robin spaces. This aspect should be commented on.

3) To which degree does particle movement paravascular within the subarachnoid space compare with tracer movement within the Virchow-Robin spaces? In humans, intrathecal injection of MRI contrast agent serving as tracer demonstrates tracer enrichment surrounding e.g. the medial cerebral artery, and tracer enriches the arteries within the subarachnoid space in antegrade direction (see Ringstad, Vatnehol and Eide, 2017, Figure 3).

4) Are the present authors assessing particle movement in the same location in mice, i.e. where the arteries reside within the subarachnoid space? Since glymphatic circulation refers to perivascular transport in VR spaces and along basement membrane, the relevance of the present study needs to be addressed.

5) Discussion, first paragraph. It is mentioned that "several published papers have hypothesized that the observed flows might instead be artifacts of non-natural elevation of ICP caused by tracer infusion into the cisterna magna." References are needed.

6) The authors might comment whether this methodology are to be used in future studies when assessing in vivo paravascular tracer movement following tracer injection to cisterna magna.

7) One main open questions of reviewer 2 is simple surprise at how close the results were to the original experiment published by Mestre et al., 2018. In the original experiment, fluid is injected at 2 microL/min, which is a significant increase above the 0.38 microL/min from natural CSF production. Any discussion on the fact that adding 500% more volumetric flow and not show any change in perivascular space flow is helpful. Where did all the extra, injected fluid go?

---

## [Author Response]

Essential revisions:1) One aspect that the authors need to clarify is the location for particle movement, which is referred to as the "surface periarterial spaces". This location should be better explained.2) Do the authors refer to branches of the medial cerebral artery within the subarachnoid space and surrounded by the arachnoid membrane?

We thank the reviewers for pointing out this need for clarification. The particles observed in these experiments (as shown in Figure 2) are localized in the perivascular spaces (PVSs) of the cortical branches of the middle cerebral artery (MCA). These PVSs are continuous with the subarachnoid space and surrounded by the pial membranes (arachnoid and pia). In the Introduction, we have added a description of this location (surface periarterial spaces) and to be consistent, we have replaced the phrase “surface periarterial spaces” throughout the revised manuscript with “perivascular spaces surrounding pial arteries”, “pial perivascular spaces” or equivalently “PVSs surrounding cortical branches of the MCA” where appropriate. Additionally, the caption describing Figure 2 has also been modified to state this location explicitly. A new sentence in the corresponding Results section now states:

“The images were acquired through a sealed cranial window using intravital two-photon microscopy. The cranial window was prepared on the right anterolateral parietal bone to visualize the cortical branches of the MCA, as chosen in previous studies.”

I would assume it is difficult to perform measurements within the Virchow-Robin spaces. This aspect should be commented on.

As the reviewer notes, we did not perform measurements within the Virchow-Robin (VR) spaces, focusing only on flows observed in the PVSs of pial arteries. Accurate particle tracking in the VR spaces is limited by the low temporal resolution of current microscopes in the third dimension (z-direction) that is required to visualize flow in these spaces. Additionally, we do not observe penetration of the microspheres along the VR spaces. As in previous studies that use fluorescent microspheres (Mestre et al., 2018; Bedussi et al., 2017), we attribute this limitation to the size of the particles used, as they are sieved when fluid enters the smaller PVSs of penetrating vessels (including VR spaces). These aspects have been commented on in the fourth paragraph of the Discussion section in the revised manuscript. We agree that the characterization of CSF flow along PVSs of penetrating vessels is a worthy topic for future study.

3) To which degree does particle movement paravascular within the subarachnoid space compare with tracer movement within the Virchow-Robin spaces? In humans, intrathecal injection of MRI contrast agent serving as tracer demonstrates tracer enrichment surrounding e.g. the medial cerebral artery, and tracer enriches the arteries within the subarachnoid space in antegrade direction (see Ringstad, Vatnehol and Eide, 2017, Figure 3).

Data collected using multiple complementary methodologies in mice, including optical and magnetic resonance imaging (MRI) (e.g., Iliff et al., 2012; Mortensen et al., 2019; Koundal et al., 2020) show evidence of flow continuing from the pial PVSs and penetrating deeper into the brain tissue. However, our quantitative analysis of CSF flow cannot be extended to the VR spaces due to particle size and technical limitations in microscopy, as noted above. We can speculate, based on the continuity equation that expresses conservation of mass, that the net flow from the PVSs of pial arteries must continue in some form through other contiguous PVSs along penetrating arteries, arterioles, capillaries, venules, etc. We have addressed issues regarding the flow in VR spaces in the fourth paragraph of the Discussion section in the revised manuscript.

Nonetheless, we want to emphasize that the focus of the current study is to demonstrate that the bulk flows of CSF observed and quantified in the PVSs of the pial vessels (Iliff et al., 2012; Xie et al., 2013; Mestre et al., 2018) are not artifacts of the tracer injection process, but are natural and have a dominant oscillatory component of flow strongly correlated to the cardiac cycle.

4) Are the present authors assessing particle movement in the same location in mice, i.e. where the arteries reside within the subarachnoid space? Since glymphatic circulation refers to perivascular transport in VR spaces and along basement membrane, the relevance of the present study needs to be addressed.

In the present study, the location of the acquired images of the tracer particles *is the same* as in our previous study (Mestre et al., 2018), now clarified in the Discussion section of the revised manuscript. Specifically, this location is the perivascular spaces of the cortical branches of the MCA near the surface of the brain and within the pial membrane. Relevance of the present study in the context of perivascular transport throughout the glymphatic circulation can be found in the Introduction and Discussion sections of the revised manuscript.

5) Discussion, first paragraph. It is mentioned that "several published papers have hypothesized that the observed flows might instead be artifacts of non-natural elevation of ICP caused by tracer infusion into the cisterna magna." References are needed.

A list of references that speculated whether injection-induced artifacts might be propelling the fluid flow was provided in the Introduction of the original version. These references and a recently published study that also speculates on injection-induced artifacts (Faghih and Sharp, 2021) are now cited again in the Discussion of the revised manuscript.

6) The authors might comment whether this methodology are to be used in future studies when assessing in vivo paravascular tracer movement following tracer injection to cisterna magna.

We do not consider the dual-syringe infusion method necessary for future studies that adopt prevalent single-injection methods to infuse tracer and study CSF flows in pial periarterial spaces. Using the dual-syringe method, we show unequivocally that flows observed in pial periarterial spaces, both in the current study and in previous reports (Iliff et al., 2012; Xie et al., 2013; Bedussi et al., 2017; Mestre et al., 2018), are affected very little or not at all by the adopted injection methodology. The dual-syringe method is useful as a validation, since it does not increase the volume of fluid in the skull nor the ICP. However, it involves implanting two needles into the cisterna magna, adding an additional step to an already complex surgery, reducing its success rate, and requiring the use of more animals. In addition, a significant portion of the injected microspheres is siphoned into the withdrawal line: we found significant differences in microsphere counts between the single-injection and dual-syringe protocols (19,728 vs. 3,100 particle tracks on average per mouse per 5 min injection). The significant siphoning renders the dual-syringe infusion method not as suitable for studies that inject fluorescent dyes to visualize in vivo flows, as opposed to fluorescent microspheres, given the reduced amounts of net tracer infusion. Recommendations on the suitability of this method are now provided in the final paragraph of the Discussion section in the revised manuscript.

7) One main open questions of reviewer 2 is simple surprise at how close the results were to the original experiment published by Mestre et al., 2018. In the original experiment, fluid is injected at 2 microL/min, which is a significant increase above the 0.38 microL/min from natural CSF production. Any discussion on the fact that adding 500% more volumetric flow and not show any change in perivascular space flow is helpful. Where did all the extra, injected fluid go?

We thank the reviewer for raising a key and interesting point. Indeed, the fact that the measured bulk flows using either injection protocol match so closely allows us to conclude that prevalent tracer infusion methods do not induce the observed flows in the perivascular spaces. We have now commented on the possible transport pathways for the extra fluid injected with the single-injection method in the fifth paragraph of the Discussion section in the revised manuscript.

Briefly, we hypothesize that the changes in volume and ICP induced by the tracer infusion at 2 μL/min likely contribute to CSF transport by transiently accelerating CSF outflow via routes such as the meningeal lymphatic vessels located around the venous sinuses and at the base of the skull, as well as cranial and spinal nerves (Louveau et al., Nature 2015; Ma et al., 2019; Ma et al., J. Exp. Med. 2019; Ahn et al., 2019). Drainage via these routes is rapid since CSF tracers can be detected in the cervical lymph nodes within minutes after injection. Attempts to quantify the glymphatic and lymphatic transport using radiolabeled tracers and contrast agents for T1 mapping techniques have estimated that around 20 % of the tracers injected into the cisterna magna is transported into the brain, implying that most of the tracer mass migrates to spine and/or drains into different efflux routes (Iliff et al., 2012; Lee et al., 2018).